# Transscleral photodynamic therapy with a chlorin e6: An experimental study of exposure parameters and therapeutic window

Ernest V. Boiko[1,2,3], Elena V. Samkovich[1]*, Irina E. Panova[1,2,3], Alexander A. Ivanov[4], Sergey B. Shevchenko[5], Sergey L. Vorobyev[6], Elizaveta S. Kalashnikova[6], Victoria G. Gvazava[1], Elizaveta A. Masian[1], Alexandra E. Kim[1]

1 S. Fyodorov "Eye Microsurgery" Federal State Institution, St. Petersburg Branch, Saint Petersburg, Russian Federation, 2 Department of Ophthalmology, Federal State Budgetary Educational Institution of Higher Professional Education «North-Western State Medical University named after I.I. Mechnikov», Saint-Petersburg, Russian Federation, 3 Department of Ophthalmology and Otolaryngology, Federal State Budgetary Educational Institution of Higher Professional Education "Saint-Petersburg State University", Saint Petersburg, Russian Federation, 4 LLC "Alcom Medica", Saint Petersburg, Russian Federation, 5 Federal State Budgetary Scientific Institution "Institute of Experimental Medicine", Saint-Peterburg, Russian Federation, 6 "National Center for Clinical Morphological Diagnostics", Saint Petersburg, Russian Federation

* e.samkovich@mail.ru

## Abstract

### Purpose

To define optimal exposure parameters and the therapeutic window for transscleral photodynamic therapy (TSPDT) with chlorin e6 by evaluating clinical, histological, and thermal effects of subthreshold, therapeutic, and suprathreshold settings in rabbit eyes.

### Methods

The study was conducted on 21 healthy rabbits. TSPDT was performed using a 660 nm laser and chlorin e6 (2.5 mg/kg). Transscleral probes (5 mm: 0.1 W, 0.17 W, 0.3 W; 10 mm: 0.3 W, 0.6 W) with integrated thermosensors were used. Enucleation and histological analysis were performed 14 days post-irradiation.

### Results

Fundus examination on day 14 revealed distinct treatment zones correlating with laser settings. The therapeutic window was defined as 0.14–0.17 W (5 mm probe; power density: 0.693–0.866 W/cm²; energy density: 415.8–519.6 J/cm²) and 0.48–0.6 W (10 mm probe; 0.611–0.764 W/cm²; 366.6–458.4 J/cm²) with 600 s exposure time, achieving selective choroidal damage without scleral or retinal injury ($\Delta T \leq 4.5°C$). Suprathreshold settings ($\geq 0.3$ W for 5 mm; $\geq 0.6$ W for 10 mm) induced retinal necrosis (up to 50%) and scleral coagulation ($\Delta T \geq 8°C$) with power densities exceeding 0.866 W/cm² (5 mm) and 0.764 W/cm² (10 mm).

**Data availability statement:** All relevant data are within the manuscript and its Supporting Information files.

**Funding:** This study was supported by a grant from the Russian Science Foundation, No. 24-75-00047, https://rscf.ru/project-24-75-00047/ Author who received award: Samkovich E. V. Sponsors and founders didn't play any role in the study design, data collection, analysis, decision to publish and preparation of the manuscript.

**Competing interests:** The authors have declared that no competing interests exist.

## Conclusion

TSPDT with chlorin e6 enables selective targeting of intraocular pathological tissues while preserving scleral and retinal integrity. Defining the therapeutic window and using real-time thermal monitoring enhances treatment safety. These findings lay a foundation for clinical protocols for uveal melanoma and other intraocular tumors.

## Introduction

Photodynamic therapy (PDT) using a 660 nm laser and chlorin-based photosensitizers (PS) selectively targets pathological tissue and offers a promising organ-preserving treatment for intraocular tumors such as uveal melanoma (UM). PDT has traditionally used a transpupillary route with first- (verteporfin) and second-generation (chlorin-based) agents [1–5]. Chlorin-based PS optimize near-infrared absorption, enhancing penetration and minimizing melanin and hemoglobin interaction [5–7]. However, transpupillary limitations in peripheral and large tumors require alternative methods. Transscleral PDT (TSPDT) addresses these challenges, with Boiko E.V. and colleagues having established transscleral delivery principles [8–10].

PDT efficacy depends on reactions between the photosensitizer, light, and molecular oxygen [11]. Temperature control is critical: exceeding 40°C destabilizes the photosensitizer, reduces singlet oxygen production by 50–70%, and induces hypoxia, shifting toward thermotherapy and coagulative necrosis [12–14]. These effects compromise selectivity, emphasizing strict laser parameter control and advanced light delivery systems.

The therapeutic window of TSPDT depends on laser settings, probe design, photosensitizer properties, and anatomical factors. Lack of standardization remains a challenge. Developing TSPDT protocols requires integrating modeling, experimental validation, thermal monitoring, and comprehensive sclera-choroid-retina complex analysis.

The aim of this study was to determine the exposure parameters and therapeutic window for TSPDT with chlorin e6 by assessing clinical, histological, and thermal effects of subthreshold, therapeutic, and suprathreshold settings in rabbit eyes.

## Materials and methods

Twenty-one rabbits (3.0–3.5 kg) underwent ophthalmic screening. Procedures followed the Declaration of Helsinki and Directive 2010/63/EU, with approval from the local Ethics Committee of the Federal State Budgetary Scientific Institution "IEM" (Protocol No. 4/24, dated October 24, 2024).

TSPDT was performed under anesthesia (1 mL Zoletil 50 mg/mL, 0.3 mL Rometar 20 mg/mL) using a 660 nm diode laser (ALOD-01, Alcom Medica, Russia) and intravenous chlorin e6 (Photoran®) at 2.5 mg/kg, administered 30 minutes before irradiation. Chlorin e6 was selected for tissue selectivity, singlet oxygen generation, and low toxicity [2,4,15,18,24]. A dose of 2.0–3.0 mg/kg and a 30-minute interval were selected based on pharmacokinetic data and rabbits' faster metabolism [16–20].

Laser energy was delivered using flat-surfaced transscleral probes (5 mm, 10 mm) with integrated thermosensors (patent No. 2025105194, RU). The probe diameters were chosen based on anticipated clinical use: the 10 mm probe enables treatment of tumors with large basal diameters by covering a broader area and reducing exposure time, while the 5 mm probe allows for precise application to smaller or peripheral lesions. Due to the rabbits' globe size, two zones were possible with the 10 mm probe and three zones with the 5 mm probe.

The laser beam exhibited non-uniform power distribution, with maximum intensity at the center and gradual attenuation toward the periphery, resulting in a ~20% difference between central and peripheral zones. This non-uniformity was factored into calculations of power density (0.382–1.528 W/cm² for the 5 mm probe; 0.382–0.764 W/cm² for the 10 mm probe) and energy density (229.2–916.7 J/cm² and 229.2–458.4 J/cm², respectively). Experimental parameter planning was informed by prior in vitro dosimetric studies demonstrating a 2.5-fold (up to 60%) reduction in irradiance through cadaveric sclera [21]. Additionally, literature-reported in vivo PDT parameters (power density: 0.3–2.0 W/cm²; energy density: 50–300 J/cm²) were reviewed [22–25]. To compensate for energy loss and ensure adequate fluence in deeper ocular layers, laser parameters were proportionally increased, resulting in higher power and energy densities than typically used in oncologic PDT [7]. Output was pre-calibrated using a power control unit (PDI-01, Alcom Medica, Russia).

Rabbits were divided into two groups: group I (5 mm probe, 10 rabbits, 20 right eyes): 0.1 W, 0.17 W, 0.3 W; exposure 600 s. group II (10 mm probe, 10 rabbits, 20 left eyes): 0.3 W, 0.6 W; exposure 600 s. One eye served as a control. Temperature was recorded at baseline, 5 min, and 10 min. ΔT was defined as the difference between baseline and 10-min readings.

Post-treatment evaluation included ophthalmic examination, fundus photography (ZEISS CLARUS 500) under mydriasis, and ultrasonography (PHILIPS Affinity 50, L15-7io, 15–7 MHz).

Histological analysis was performed 14 days after TSPDT. Eyes were fixed in 10% formalin, processed (Sakura VIP6), paraffin-embedded, sectioned at 4 µm (Sakura SRM200), and stained with hematoxylin and eosin (Sakura Tissue Tek Prisma Film).

## Statistical analysis

Statistical analysis was performed using SPSS software. Quantitative variables (temperature, power) were analyzed by Student's t-test and ANOVA; categorical variables (damage presence) by $\chi^2$ test. A p-value < 0.05 was considered significant.

## Results

Clinical examination revealed localized reactions at irradiation sites. Within 24 hours, mild conjunctival injection and pinpoint hemorrhages were observed, without signs of pain. Fundus examination on day 14 showed distinct lesions corresponding to laser settings. In eyes treated with the 5 mm probe: subthreshold (0.1 W), minimal choroidal atrophy and edema; therapeutic (0.17 W), pronounced atrophy with sharp borders and swelling; suprathreshold (0.3 W): extensive atrophy, marked edema, and depigmentation (Fig 1A). In eyes treated with the 10 mm probe (0.3 W, 0.6 W), two larger foci with similar changes were observed (Fig 1B). The control eye showed no pathological changes.

As shown in Table 1, histological evaluation of Group I (5 mm probe, three exposure settings) demonstrated a progressive pattern of tissue alterations with increasing laser power: at the subthreshold setting (0.1 W), no structural changes were observed in the sclera. In the choroid, mild edema and thrombosis occurred in 15% of cases (3/20) (Fig 2A). At the therapeutic setting (0.17 W), selective choroidal damage was observed, with thrombosis in 65% (13/20, p < 0.001) and vascular ectasia in 45% (9/20). The sclera and retina remained unaffected, with no scleral homogenization and mild retinal edema in 5% (1/20) (Fig 2B). At the suprathreshold setting (0.3 W), damage was more extensive, with scleral homogenization in 30% (6/20, p = 0.001), retinal necrosis in 35% (7/20, p < 0.001), and choroidal hemorrhage in 40% (8/20).

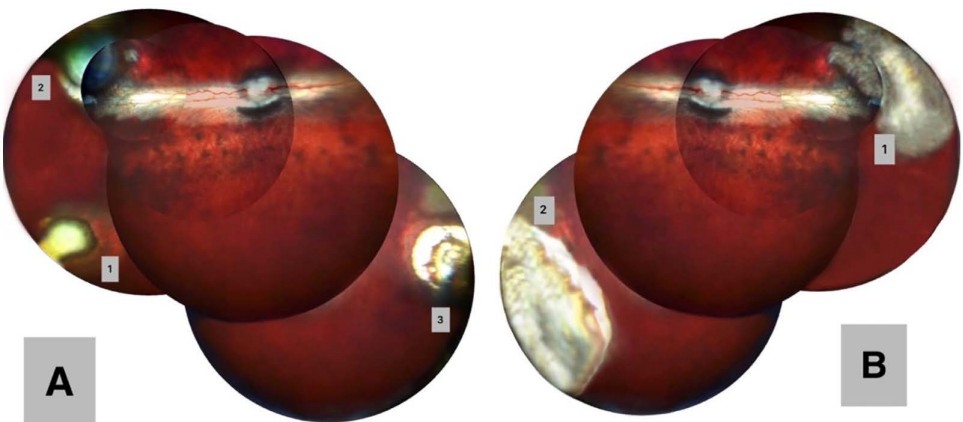

**Fig 1. Fundus photographs of rabbit eyes on day 14 after TSPDT: (A) Right eye – three treatment zones corresponding to different laser power settings: zone 1 - 0.1 W, zone 2 - 0.17 W, zone 3 - 0.3 W. (B) Left eye – two treatment zones: zone 1 - 0.3 W, zone 2 - 0.6 W.**

**Table 1. Group I: TSPDT with 5 mm Transscleral Probe (three power settings). Histological changes in the sclera-choroid-retina complex.**

| Parameter | Subthreshold (0.1 W) | Therapeutic (0.17 W) | Suprathreshold (0.3 W) | p-value |
|---|---|---|---|---|
| **Sclera** | | | | |
| Homogenization | 0% (0/20) | 0% (0/20) | 30% (6/20) | p = 0.001 |
| Sclerosis | 0% (0/20) | 0% (0/20) | 20% (4/20) | p = 0.02 |
| **Choroid** | | | | |
| Thrombosis | 15% (3/20) | 65% (13/20)[a] | 90% (18/20)[b] | p < 0.001 |
| Vascular ectasia | 10% (2/20) | 45% (9/20)[a] | 75% (15/20)[b] | p < 0.001 |
| Hemorrhage | 0% (0/20) | 10% (2/20) | 40% (8/20)[b] | p = 0.005 |
| **Retina** | | | | |
| Edema | 0% (0/20) | 5% (1/20) | 20% (4/20)[a] | p = 0.01 |
| Necrosis | 0% (0/20) | 0% (0/20) | 35% (7/20)[b] | p < 0.001 |

[a] p < 0.05 vs. subthreshold.

[b] p < 0.01 vs. therapeutic setting.

Table 2 summarizes histological changes in Group II (10 mm probe), where a similar dose-dependent pattern was observed: increasing laser power correlated with progressive ocular tissue alterations, with statistically significant differences between exposure settings.

As shown in Table 2, histological analysis of Group II (10 mm probe, two exposure settings) revealed a dose-dependent progression of ocular tissue changes, with statistically significant differences between exposure levels. At the therapeutic setting (0.3 W), mild scleral homogenization was observed in 10% of cases (2/20), while choroidal thrombosis was present in 85% (17/20, p < 0.001). The retinal architecture was preserved in the majority of eyes (85%), with retinal necrosis detected in only 10% (2/20). At the suprathreshold setting (0.6 W), tissue damage was more pronounced. Scleral homogenization increased to 30% (6/20, p = 0.03), retinal necrosis was observed in 50% of cases (10/20, p < 0.001), and choroidal hemorrhages occurred in 55% (11/20, p = 0.002) (Fig 3A, Fig 3B).

Histological examination of the control eye (no TSPDT performed) revealed no pathological alterations: the sclera, choroid, and retina retained normal architecture with no evidence of homogenization, sclerosis, thrombosis, or necrosis.

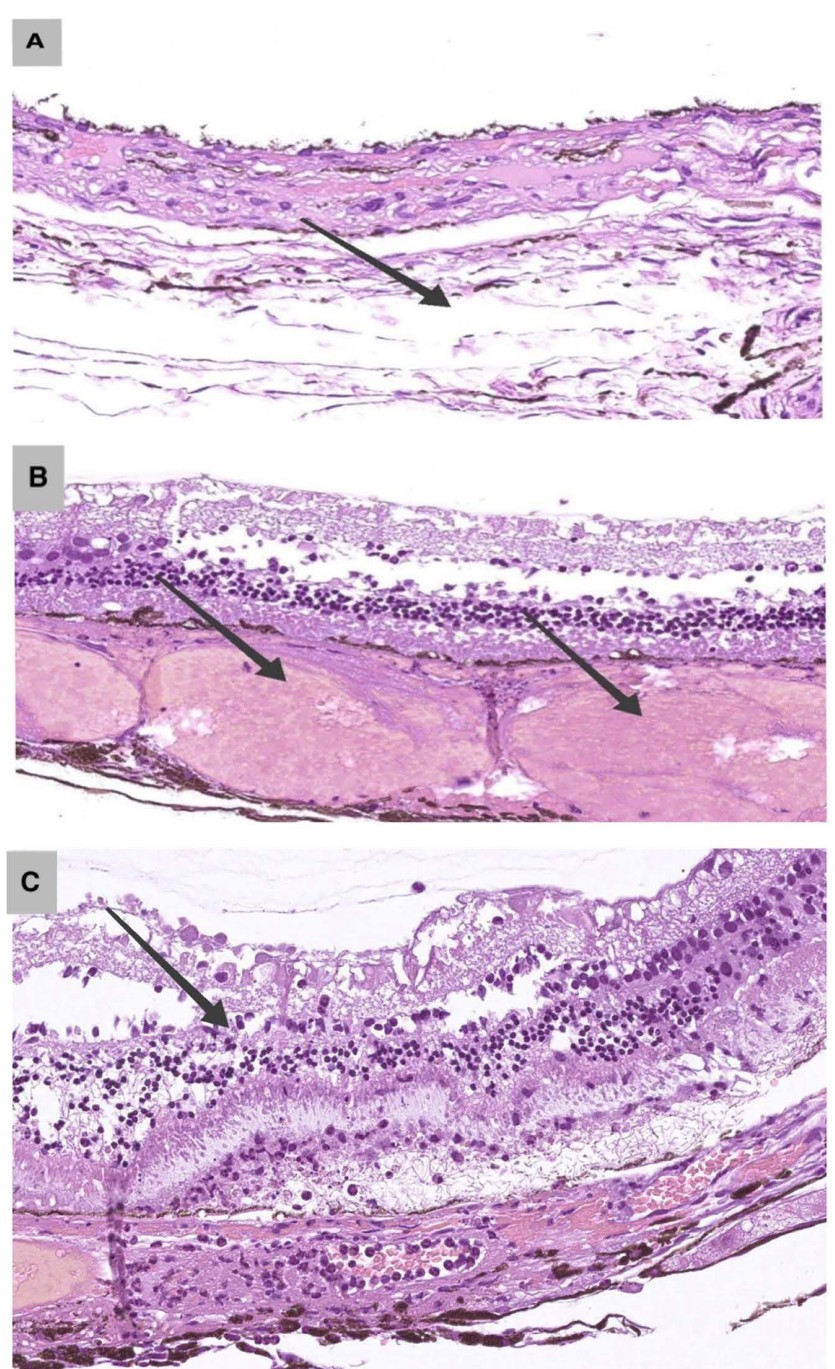

**Fig 2. A. Histological changes in rabbit ocular tissues under subthreshold parameters (0.1 W).** Stromal edema of the choroid (arrow). H&E, ×400. 5 mm probe. **B.** Histological changes in rabbit ocular tissues under therapeutic parameters (0.17 W). Choroidal vascular ectasia, erythrocyte thrombi (arrows), and erythrocyte extravasation. H&E, ×400. 5 mm probe.

**Table 2. Group II: TSPDT with 10 mm transscleral probe (two power settings). Histological changes in the sclera-choroid-retina complex.**

| Parameter | Therapeutic (0.3 W) | Suprathreshold (0.6 W) | p-value |
|---|---|---|---|
| **Sclera** | | | |
| Homogenization | 10% (2/20) | 30% (6/20) | p = 0.03 |
| Sclerosis | 0% (0/20) | 20% (4/20) | p = 0.01 |
| **Choroid** | | | |
| Thrombosis | 85% (17/20) | 95% (19/20) | p = 0.1 |
| Vascular ectasia | 70% (14/20) | 85% (17/20) | p = 0.05 |
| Hemorrhage | 25% (5/20) | 55% (11/20) | p = 0.002 |
| **Retina** | | | |
| Edema | 15% (3/20) | 45% (9/20) | p = 0.008 |
| Necrosis | 10% (2/20) | 50% (10/20) | p < 0.001 |

The dynamics of scleral temperature changes (ΔT) under various TSPDT settings are presented in Table 3.

Temperature data obtained using probes with integrated thermosensors revealed a clear dose-dependent relationship between laser power and scleral heating. As shown in Table 3, use of the 5 mm probe at increasing power levels (from 0.1 W to 0.3 W) led to a progressive rise in ΔT from 2.8°C to 8.6°C (p < 0.001), which correlated with a higher incidence of retinal necrosis and scleral homogenization. A similar trend was observed with the 10 mm probe: increasing the power from 0.3 W to 0.6 W resulted in a ΔT rise from 3.7°C to 7.8°C (p < 0.001), coinciding with increased rates of choroidal hemorrhage (from 25% to 55%) and retinal necrosis (from 10% to 50%).

## Discussion

Although transpupillary PDT is clinically established, the transscleral approach remains largely unexplored. This preclinical study shows that TSPDT with chlorin e6 can effectively and safely treat intraocular tumors. The therapeutic window and key parameters (laser power, power density, energy density, thermal dynamics) were first-ever systematically defined by correlating clinical, histological, and thermal responses in a controlled model. Although conducted in healthy eyes, these findings provide a basis for safety evaluation before tumor studies and clinical translation.

The results highlight the importance of precise laser exposure control, including beam non-uniformity and the mismatch between flat probe tips and the curved scleral surface. A ~20% power drop from center to periphery, though clinically acceptable, requires careful dosimetry. Anatomical factors such as scleral thickness, pigmentation, and vascular density may also influence energy distribution and thermal risk [10,26].

Thermal monitoring is crucial for safety and effective photochemical activation [14,27]. PDT relies on reactive oxygen species generated by light, photosensitizer, and molecular oxygen Without temperature control, prior studies reported overheating and reduced photochemical efficiency, shifting treatment toward coagulative necrosis [11,12]. In this study, maintaining ΔT ≤ 4.5°C (0.17 W for 5 mm; 0.3 W for 10 mm probes) minimized damage and preserved chlorin e6 activity. Exposure parameters were adjusted to compensate for light attenuation and anatomical variability. While three power levels were tested, the therapeutic window can be adapted based on tumor location and scleral thickness. Subthreshold settings (0.1 W, 5 mm) caused no damage; therapeutic settings (0.17 W, 5 mm; 0.3 W, 10 mm) selectively targeted the choroid; suprathreshold settings (≥0.3 W, 5 mm; ≥0.6 W, 10 mm) induced thermal injury.

Despite its promise, PDT remains limited by penetration depth in intraocular tumors. Scattering and reflection in tissues like sclera and skin reduce fluence at depth. Similar issues affect photothermal therapies [28]. Photosensitizer activation spans 405–900 nm, with penetration depth depending on wavelength and tissue optics [29]. First-generation

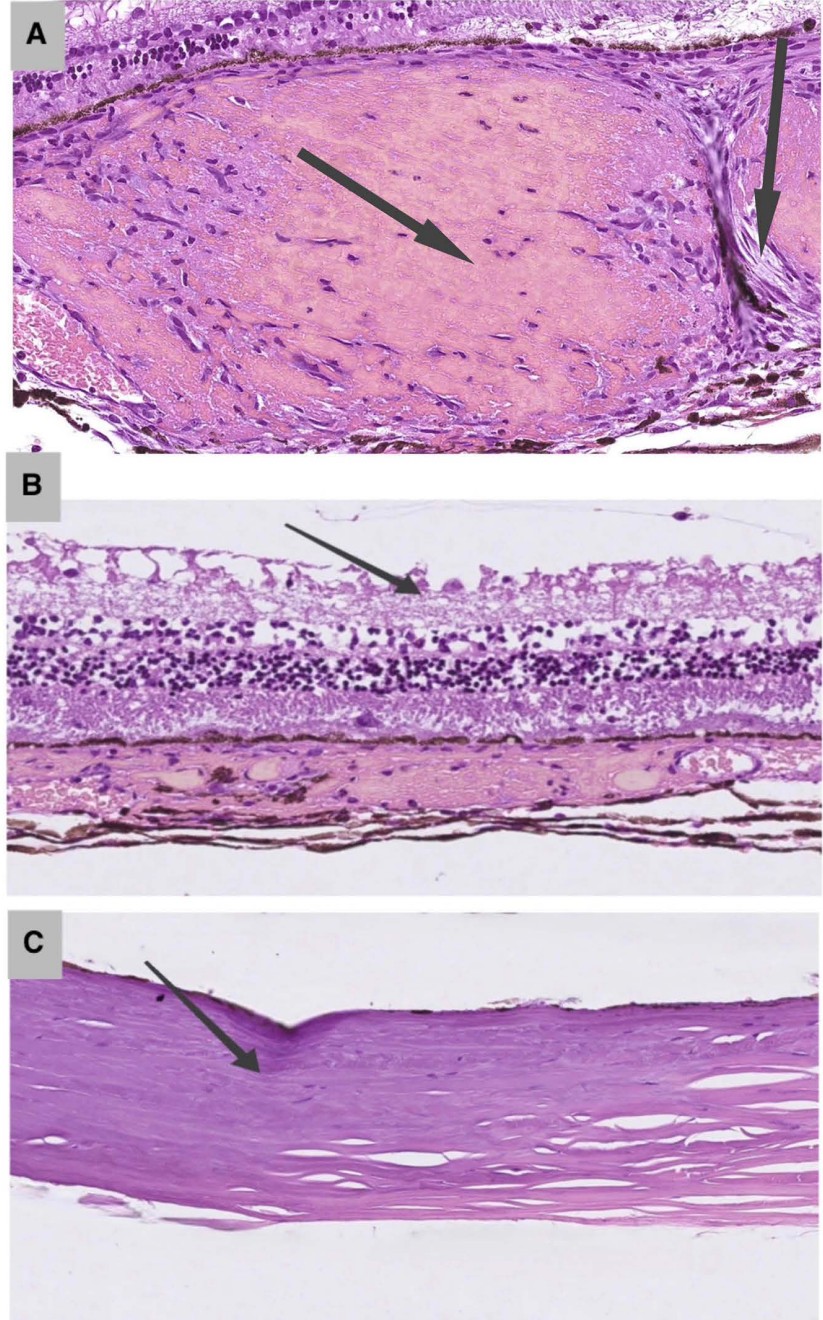

**Fig 3. A. Histological changes in rabbit ocular tissues under suprathreshold parameters (0.6 W).** Destructive retinal alterations, including ganglion cell necrosis (arrow), and choroidal vascular thrombosis. H&E, ×200. 10 mm probe. **B.** Histological changes in rabbit ocular tissues under suprathreshold parameters (0.6 W). Thermal damage to the sclera, characterized by homogenization of fibroblastic fibers (arrow). H&E, ×200. 10 mm probe.

agents (hematoporphyrin derivatives, 5-aminolevulinic acid), absorbing in the 400–630 nm range, are strongly absorbed by superficial tissues, limiting penetration to 1–3 mm [30,31]. Thus, superficial tissues absorb much of the light, preventing effective delivery to intraocular tumors like UM. Chlorin-based photosensitizers, activated at 660 nm, penetrate

**Table 3. Dynamics of scleral temperature changes (ΔT) during TSPDT using 5 mm and 10 mm transscleral probes.**

| Setting | T at baseline (°C) | T at 5 min (°C) | T at 10 min (°C) | ΔT (°C) | p-value |
|---|---|---|---|---|---|
| **5 mm Transscleral Probe** | | | | | |
| **0.1 W** | 35.2±0.5 | 37.1±0.7 | 38.0±0.8 | 2.8 | |
| **0.17 W** | 36.0±0.6 | 39.5±0.9[a] | 40.2±1.0[a] | 4.2 | p<0,001 (vs 0,1 W) |
| **0.3 W** | 36.5±0.7 | 42.3±1.2[b] | 45.1±1.5[b] | 8.6 | p<0,001 (vs 0,17 W) |
| **10 mm Transscleral Probe** | | | | | |
| **0.3 W** | 37.8±0.8 | 40.1±1.0 | 41.5±1.1 | 3.7 | p<0,001 |
| **0.6 W** | 38.2±0.9 | 43.5±1.3[b] | 46.0±1.6[b] | 7.8 | p<0,001 |

[a] p<0.05 vs. subthreshold.

[b] p<0.01 vs. therapeutic setting.

3.5–4.4 mm, aided by reduced scattering and lower melanin and hemoglobin interaction [5,6]. Attenuation depth – where light intensity drops to 37% - depends on absorption, scattering, and reflection [32]. For wavelengths near 660 nm, attenuation depths of 3–5 mm match clinical outcomes with chlorin e6 [2,9,21]. Chlorin e6 also offers high pathological selectivity and low systemic toxicity, improving the therapeutic index [7,31].

TSPDT offers several advantages over conventional methods for UM, including transpupillary thermotherapy (TTT) and Ruthenium-106 plaque brachytherapy (BT) [33–35]. Key benefits include access to peripheral lesions inaccessible via transpupillary approaches, the ability to combine probes of different diameters (e.g., 5 mm for peripheral foci; 10 mm for central tumors) and integrated thermometry for real-time feedback and thermal safety. Unlike BT, which uses radioactive sources, requires applicator removal, and carries postradiation complications, TSPDT is a non-radioactive, single-step procedure [33–36]. It can also complement transpupillary PDT and TTT for bilateral tumor targeting [9].

This study has limitations. The healthy eye model does not fully simulate tumor microenvironments, including neovascularization and optical heterogeneity. Moreover, the non-uniform distribution of laser power, combined with the flat design of the probe applied to the spherical surface of the eyeball, may exacerbate energy losses due to light scattering and reflection. Nonetheless, this study defines TSPDT parameters (power, power density, energy density, exposure time) and supports its safety within specific parameters, laying the foundation for future intraocular tumor research and clinical protocols. Future directions include integrating closed-loop thermal feedback, developing curved probes for better ocular fit, and reducing power variation across the spot to <10%. Additionally, new photosensitizers with absorption peaks beyond 700 nm (700–1100 nm) could enhance tissue penetration.

## Conclusions

1. The therapeutic window for TSPDT with chlorin e6 was defined as: 5 mm probe: 0.14–0.17 W (power density: 0.693–0.866 W/cm²; energy density: 415.8–519.6 J/cm²; exposure time: 600 s). 10 mm probe: 0.48–0.6 W (power density: 0.611–0.764 W/cm²; energy density: 366.6–458.4 J/cm²; exposure time: 600 s) These settings produced selective choroidal damage without thermal injury to the sclera or retina under experimental conditions.

2. Suprathreshold settings (≥0.3 W for 5 mm; ≥0.6 W for 10 mm) caused excessive heating (ΔT ≥ 8°C), leading to retinal necrosis (up to 50% of cases) and scleral coagulative damage. Corresponding power and energy densities exceeded >0.866 W/cm² and >519.6 J/cm² (5 mm probe) and >0.764 W/cm² and >458 J/cm² (10 mm probe).

3. Thermal monitoring via innovative probes with integrated sensors enhances TSPDT safety. Maintaining ΔT ≤ 4.5°C is recommended to minimize thermal risk.

4. The obtained results may serve as a foundation for the development of standardized TSPDT protocols for UM and other intraocular tumors.

## Supporting information

**S1 Table. Complete experimental dataset.** This file contains the full dataset underlying all study findings, including detailed records of scleral temperature changes (ΔT) and histological assessment scores for all rabbit eyes treated with 5 mm and 10 mm transscleral probes under subthreshold, therapeutic, and suprathreshold laser settings.
(XLSX)

## Author contributions

**Conceptualization:** Ernest V. Boiko, Elena Vladislavovna Samkovich, Irina E. Panova.

**Data curation:** Ernest V. Boiko, Elena Vladislavovna Samkovich, Irina E. Panova, Sergey L. Vorobyev.

**Formal analysis:** Elena Vladislavovna Samkovich, Sergey L. Vorobyev, Elizaveta S. Kalashnikova, Victoria G. Gvazava, Elizaveta A. Masian.

**Funding acquisition:** Elena Vladislavovna Samkovich.

**Investigation:** Ernest V. Boiko, Elena Vladislavovna Samkovich, Irina E. Panova, Sergey L. Vorobyev, Elizaveta S. Kalashnikova, Victoria G. Gvazava, Elizaveta A. Masian.

**Methodology:** Ernest V. Boiko, Elena Vladislavovna Samkovich, Irina E. Panova, Sergey L. Vorobyev.

**Project administration:** Ernest V. Boiko, Elena Vladislavovna Samkovich, Sergey B. Shevchenko.

**Resources:** Ernest V. Boiko, Alexander A. Ivanov, Sergey B. Shevchenko, Sergey L. Vorobyev.

**Supervision:** Ernest V. Boiko.

**Validation:** Ernest V. Boiko, Elena Vladislavovna Samkovich, Irina E. Panova, Sergey L. Vorobyev.

**Writing – original draft:** Elena Vladislavovna Samkovich, Elizaveta S. Kalashnikova, Victoria G. Gvazava, Elizaveta A. Masian.

**Writing – review & editing:** Elena Vladislavovna Samkovich, Alexander A. Ivanov, Victoria G. Gvazava, Elizaveta A. Masian, Alexandra E. Kim.

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
