## [Decision Letter · Decision Letter 0]

21 Oct 2025

Dear Dr. Samkovich,

Thank you for submitting your manuscript to PLOS ONE. After careful consideration, we feel that it has merit but does not fully meet PLOS ONE’s publication criteria as it currently stands. Therefore, we invite you to submit a revised version of the manuscript that addresses the points raised during the review process.

We look forward to receiving your revised manuscript.

Kind regards,

Vibhuti Agrahari

Academic Editor

PLOS ONE

Journal Requirements:

3. We note that your Data Availability Statement is currently as follows: “All relevant data are within the manuscript and its Supporting Information files.”

Reviewers' comments:

Reviewer's Responses to Questions

**Comments to the Author**

1. Is the manuscript technically sound, and do the data support the conclusions?

Reviewer #1: Partly

Reviewer #2: Yes

2. Has the statistical analysis been performed appropriately and rigorously?

Reviewer #1: Yes

Reviewer #2: Yes

3. Have the authors made all data underlying the findings in their manuscript fully available?

Reviewer #1: No

Reviewer #2: Yes

4. Is the manuscript presented in an intelligible fashion and written in standard English?

Reviewer #1: No

Reviewer #2: Yes

Reviewer #1: Comments and Queries from Reviewer

The topic of article presented is appreciative and study reports certain worthwhile information that can be considered useful for clinical translation in future. Data appears detailed and thorough. The author appears to have an appreciative effort in discussion section to provide thorough arguments regarding the study as well as its drawback and future prospects in discussion. Thus, it is a useful study to be reported. Although, there are a few suggestions and queries for the author, which are as follows:

1. https://journals.sbmu.ac.ir/jlms/article/view/47901/35787 (Transscleral Photodynamic Therapy with a Chlorin e6 Photosensitizer in a Rabbit Experimental Model of an Intraocular Mass Lesion: TS-PDT in experimental ocular mass lesion model).

a. The reviewer suggests the author to explain if this study is different from the given manuscript. If possible please share the manuscript PDF, as it is not accessible.

2. In the Line 179-181 (The therapeutic window and key parameters (laser power, power density, energy density, thermal dynamics) were first-ever systematically defined by correlating clinical, histological, and thermal responses in a controlled model)

a. Is the author sure about the claims mentioned above? Please also shed some light and explain about the clinical correlation other wise the author may modify the statement.

3. Twenty-one rabbits (3.0–3.5 kg) underwent ophthalmic screening. Procedures followed the 69 Declaration of Helsinki and Directive 2010/63/EU, with approval from the local Ethics Committee 70 of the Federal State Budgetary Scientific Institution “IEM” (Protocol No. 4/24, dated October 24, 71 2024).

a. This paragraph and like this other major content of the article has come under plagiarism indicating use of some AI software. The reviewer advise to the author to kindly re-check whole manuscript for the AI plagiarism and paraphrase the text.

4. The manuscript needs to review by author for reference citation correction (for example reference 5 or 6 doesn’t suggest correct citation for line 205, and First sentence specific reference needed instead of adding reference after paras.) likewise there are more. The references are at certain places not accurate or wrongly added.

5. Material method 2nd para lacks sentences wise specific references. Rather bunch of references added altogether after couple of sentences. This confuses readers to choose correct reference for specific context.

6. The reviewer suggests the author to kindly cite references next to each sentence where it is referred to, instead of citing a bunch of references after 2-3 sentences combined. It is only confusing the readers and not giving accurate information.

7. Introduction section lacks streamlined discussions rather puts useful information in random order in a way that the discussion topics abruptly shifts from one to another without any order. The author wants to share useful information but it needs to streamlined and described in orderly manner while connecting the dots as a storyline discussion.

8. Which species of rabbits selected for animal study?

9. This sentence is unclear. This sentence has missing to write “treatment Zones”.

a. Due to the rabbits’ globe size, two zones were possible with the 10 mm probe and three zones with the 5 mm probe.

10. The following sentences appears to be unclear. How 10 Rabbits have 20 right eyes or other 10 rabbits have 20 left eyes.

a. Rabbits were divided into two groups:

Group I (5 mm probe, 10 rabbits, 20 right eyes): 0.1 W, 0.17 W, 0.3 W; exposure 600 s. Group II (10 mm probe, 10 rabbits, 20 left eyes): : 0.3 W, 0.6 W; exposure 600 s.

11. Is the author sure to use the term “Clinical” in the abstract under the purpose section for “animal-based study”? if yes, please justify.

12. Among the three zones of used in the study using 5 mm probe, two (0.1 W, and 0.17 W) are discussed although, the discussion on zone 3 (0.3 W) appears to be lacking.

13. Histological changes on 10 mm probe of 0.3 W in fig 2C is missing.

14. Please explain in line 148 author mentioned the p value as p< 0.001, however, in the table 2 group 2 it is shown as p= 0.1, which one is correct?

15. The captions for fig 3A and Fig 3B are same!?

16. Table 3 column 5 heading show T10 – T0, but T5 – T0 is not added in the table, please explain the reason. What is the clinical significance of Scleral Temperature Changes? Please answer both queries separately.

Reviewer #2: This manuscript investigates transcleral photodynamic therapy with chlorine e6 in rabbits, focusing on the parameters of exposition and therapeutic window. Overall, the manuscript is well written, with a structured abstract and a concise and clear Introduction. The Material and Methods included the information on laser parameters, as expected, besides characteristics of transscleral, and chlorine e6 dosages, allowing the reproduction of the study.

Actually, I have a doubt on this section. At page 4, the authors describe that "Rabbits were divided into two groups: group I (5 mm probe, 10 rabbits, 20 right eyes): 0.197 W, 0.17 W, 0.3 W; exposure 600 s. group II (10 mm probe, 10 rabbits, 20 left eyes): 0.3 W, 0.6 W; exposure 600 s. One eye served as a control. Temperature was recorded at baseline, 5 min, and 10 min. ΔT was defined as the difference between baseline and 10-min readings."

The question is: how can 10 rabbits have 20 right eyes or 20 left eyes? I believe this is not correct. Please review/clarify. Tables legends must describe the statistical tests used.

The main limitations of the study are the use of healthy rabbit's eye to simulate the application of therapy aimed at tumor treatment. The model does not simulate the tumor microenvironment and a deeper discussion on the possible clinical application of the methodology presented in a healthy animal model should be provided. A deeper discussion on the main challenges of the study for a clinical application should be provided.

**Do you want your identity to be public for this peer review?** For information about this choice, including consent withdrawal, please see our Privacy Policy

Reviewer #1: No

Reviewer #2: No

---

## [Author Response · Author response to Decision Letter 1]

1 Dec 2025

Dear PLOS ONE Editorial Team,

Thank you for the opportunity to submit a revised version of our manuscript PONE-D-25-24966 "Transscleral Photodynamic Therapy with a Chlorin e6: an Experimental Study of Exposure Parameters and Therapeutic Window" for consideration at PLOS ONE.

We highly appreciate the time and effort the reviewers have dedicated to analyzing our work. Their constructive comments and valuable suggestions were extremely helpful in improving our study. We have carefully analyzed all feedback and made corresponding corrections to the manuscript.

In accordance with the journal's policy, we have uploaded all required files via the Editorial Manager submission system. For your convenience and to ensure redundancy, we are also sending these files as attachments to this email:

1. The 'Response to Reviewer 1' file: Detailed point-by-point responses to all comments from Reviewer #1 in a table format.

2. The 'Response to Reviewer 2' file: Detailed point-by-point responses to all comments from Reviewer #2 in a table format.

3. The 'Revised Manuscript' file: The corrected and edited version of the manuscript. To facilitate the review process, all changes made have been highlighted in yellow within the text.

4. The requested article: As requested by Reviewer #1, we are also attaching the PDF file of our previous article (26-jlms_47901-25-R1.pdf) for the convenience of the editorial office and reviewers.

We have also addressed all additional journal requirements listed in your letter.

This work was supported by the Russian Science Foundation (RSF), grant No. 24-75-00047, https://rscf.ru/project/24-75-00047/. We kindly ask that the funding information is accurately reflected upon publication, as this is important for our reporting obligations to the funding agency.

We believe the manuscript has been significantly strengthened by the revisions made and hope it now meets the high standards for publication in PLOS ONE.

Thank you for your attention and consideration of our work.

Sincerely,

Elena Samkovich,

Ophthalmologist, Oncologist, Head of the Research and Education Department,

PhD in Medical Sciences.

Email: e.samkovich@mail.ru

---

## [Decision Letter · Decision Letter 1]

2 Jan 2026

Transscleral Photodynamic Therapy with a Chlorin e6: an Experimental Study of Exposure Parameters and Therapeutic Window

PONE-D-25-24966R1

Dear Dr. Samkovich,

We’re pleased to inform you that your manuscript has been judged scientifically suitable for publication and will be formally accepted for publication once it meets all outstanding technical requirements.

Kind regards,

Vibhuti Agrahari

Academic Editor

PLOS One

Additional Editor Comments (optional):

Reviewers' comments:

Reviewer's Responses to Questions

**Comments to the Author**

Reviewer #1: All comments have been addressed

Reviewer #2: All comments have been addressed

2. Is the manuscript technically sound, and do the data support the conclusions?

Reviewer #1: Yes

Reviewer #2: Yes

3. Has the statistical analysis been performed appropriately and rigorously?

Reviewer #1: Yes

Reviewer #2: Yes

4. Have the authors made all data underlying the findings in their manuscript fully available?

Reviewer #1: Yes

Reviewer #2: Yes

5. Is the manuscript presented in an intelligible fashion and written in standard English?

Reviewer #1: Yes

Reviewer #2: Yes

Reviewer #1: (No Response)

Reviewer #2: After the first review round, the authors have correctly addressed all the flaws pointed out by the reviewers. In particular, I am delighted with the improvement in the manuscript quality achieved by the substantial changes made to the original text, as well as with the detailed answers to the reviewers' comments that the authors have provided, which justified the methodology and clarified my doubts. I don't have any other concerns to list.

**Do you want your identity to be public for this peer review?** For information about this choice, including consent withdrawal, please see our Privacy Policy

Reviewer #1: No

Reviewer #2: No
